# Vitamin D-Related Genes and Thyroid Cancer—A Systematic Review

**DOI:** 10.3390/ijms232113661

**Published:** 2022-11-07

**Authors:** Adam Maciejewski, Katarzyna Lacka

**Affiliations:** Department of Endocrinology, Metabolism and Internal Diseases, Poznan University of Medical Sciences, 60355 Poznan, Poland

**Keywords:** vitamin D, thyroid cancer, vitamin D receptor, polymorphism, vitamin D-related genes, hydroxylase

## Abstract

Vitamin D, formerly known for its role in calcium-phosphorus homeostasis, was shown to exert a broad influence on immunity and on differentiation and proliferation processes in the last few years. In the field of endocrinology, there is proof of the potential role of vitamin D and vitamin D-related genes in the pathogenesis of thyroid cancer—the most prevalent endocrine malignancy. Therefore, the study aimed to systematically review the publications on the association between vitamin D-related gene variants (polymorphisms, mutations, etc.) and thyroid cancer. PubMed, EMBASE, Scopus, and Web of Science electronic databases were searched for relevant studies. A total of ten studies were found that met the inclusion criteria. Six vitamin D-related genes were analyzed (*VDR*—vitamin D receptor, *CYP2R1*—cytochrome P450 family 2 subfamily R member 1, *CYP24A1*—cytochrome P450 family 24 subfamily A member 1, *CYP27B1*—cytochrome P450 family 27 subfamily B member 1, *DHCR7*—7-dehydrocholesterol reductase and *CUBN*—cubilin). Moreover, a meta-analysis was conducted to summarize the data from the studies on *VDR* polymorphisms (rs2228570/*Fok*I, rs1544410/*Bsm*I, rs7975232/*Apa*I and rs731236/*Taq*I). Some associations between thyroid cancer risk (*VDR*, *CYP24A1*, *DHCR7*) or the clinical course of the disease (*VDR*) and vitamin D-related gene polymorphisms were described in the literature. However, these results seem inconclusive and need validation. A meta-analysis of the five studies of common *VDR* polymorphisms did not confirm their association with increased susceptibility to differentiated thyroid cancer. Further efforts are necessary to improve our understanding of thyroid cancer pathogenesis and implement targeted therapies for refractory cases.

## 1. Introduction

Thyroid nodules are commonly found with a prevalence as high as 65–70%, although most are benign [1,2]. Among thyroid gland malignancies (7–15% of all thyroid nodules), thyroid cancer predominates [1]. Others, such as primary thyroid lymphoma or sarcoma, are extremely rare [3]. Differentiated thyroid cancer (DTC), with its main subtypes, papillary (PTC) and follicular (FTC), is the most frequent type and accounts for over 95% of cases. Other thyroid cancer types include medullary thyroid cancer (MTC; 1–2%) and anaplastic thyroid cancer (ATC; less than 1%) [4]. Although not the top of the list of all malignancies, thyroid cancer is the most prevalent endocrine malignant neoplasm, and its incidence is observed to be rising [5]. In the case of DTC, there is generally a good prognosis with a low mortality. However, prevention, optimal management, and identification of cases with an unsatisfactory outcome risk are still challenging [6]. Instead, MTC and especially ATC are associated with poor prognosis, despite advances in medical therapies [3,6]. Therefore, further effort is needed to understand thyroid cancer pathogenesis.

Vitamin D has been the focus of interest in a broad range of disciplines in recent years. Its role is known to be far beyond the earlier perception, as vitamin D serves as an immune system regulator and alters cell differentiation and proliferation [7]. Therefore, vitamin D seems to be an excellent candidate as an environmental contributor to various autoimmune, inflammatory, and neoplastic diseases [8,9]. An inverse relationship between vitamin D concentration and cancer risk has been confirmed in the case of colon, breast, prostate, gastric, and other cancer types [10]. Moreover, there are indications of a causal relationship between decreased vitamin D level and poor cancer prognosis [11]. Active vitamin D metabolites promote cell differentiation, suppresses proliferation, and alters apoptosis and autophagy. Therefore, different stages of tumorigenesis can be affected [10].

Vitamin D supplementation seems beneficial in the case of autoimmune thyroid diseases [12]. There is also a growing amount of evidence supporting the need for vitamin D intake in the context of thyroid neoplasms. Two recent meta-analyses by Hu et al. and Zhao et al. concluded that 25(OH)D deficiency is associated with a higher thyroid cancer risk, and patients with such a diagnosis are observed to have a significantly lower 25(OH)D concentration [13,14]. Some researchers confirmed the correlation between the vitamin D level and selected clinical features of thyroid cancer, the stage of the disease, or prognosis [15,16,17]. Zhang et al. found that active vitamin D metabolites reduce proliferation and induce apoptosis of PTC cells [18]. In another in vitro study, Peng et al. showed that calcitriol is able to induce differentiation of anaplastic thyroid cancer cells [19]. Calcitriol pretreatment was suggested to improve the effectiveness of chemotherapy (adriamycin) in the course of anaplastic thyroid cancer [20]. In the course of iodine refractory PTC, vitamin D metabolites in combination with other antitumorigenic agents have been used (e.g., sorafenib, doxorubicin, paclitaxel, SAHA) to increase the effectiveness or reduce the dose of the drug needed [21,22,23]. Among the possible pathways mediating vitamin D action on DTC cells there are SIRT1-FOXO3a, Ras-MEK-ERK, PI3K/Akt, PTPN2/p-STAT3, Wnt/β-catenin axes, and some other mechanisms (Figure 1) [23,24,25,26,27,28,29]. Effective vitamin D signaling is dependent on its metabolization to the active form and further interaction with vitamin D receptor (VDR). There are also mechanisms of vitamin D enzymatic deactivation. These processes require the contribution of different molecules (including cytochrome P450 enzymes or vitamin D binding protein) encoded by genes that together can be classified as vitamin D-related genes (Figure 2) [30].

Therefore, it is expected that not only the 25(OH)D level but also vitamin D-related gene variants (polymorphisms or mutations) can modify the risk and outcome of neoplasms, including thyroid cancer. This systematic review would try to give an answer about the role of the vitamin D-related gene variants on thyroid cancer pathogenesis. In the case of a sufficient number of studies available, a meta-analysis of the data would be performed.

## 2. Methods

### 2.1. Systematic Review

The authors systematically searched the PubMed, EMBASE, Scopus, and Web of Science databases to identify evidence of an association between *VDR* or other vitamin D-related genes and thyroid cancer. The search terms were the following: vitamin D, thyroid cancer, and words describing genetic variants (polymorphism, mutation, variant), with the final query: ((vitamin D) OR (25-hydroxyvitamin D)) AND (thyroid cancer) AND ((polymorphism) OR (polymorphisms) OR (mutation) OR (variant)). Articles in English or, optionally, Polish language published prior to September 2022 were considered. The references of selected articles were checked for additional eligible studies. The systematic review was performed following the PRISMA guidelines [31].

Bibliographic research and study selection were performed independently by the two authors. In the event of any doubts or disagreement, the problem was discussed by the authors to reach the final decision. The full text of selected studies was then analyzed independently by both authors for agreement with inclusion criteria: (1) population-based or observational studies, (2) patients diagnosed with thyroid cancer (differentiated, anaplastic, or medullary), (3) control group of healthy volunteers, (4) the assessment of vitamin D-related gene variants (polymorphism, mutation, other variants). Reviews, case reports or case series, and animal or experimental studies were excluded. After eligible studies had been selected, the data of interest were extracted.

### 2.2. Meta-Analysis

A meta-analysis was performed if at least three different studies assessed the specific vitamin D-related gene variant. Access to raw data on genotype and/or allele frequencies was an additional criterion of inclusion in that case. Statistical analyses were performed using PQStat v.1.8.4.130. To assess the control groups of different studies for deviations from the Hardy–Weinberg equilibrium (HWE), the chi-square test was used. Pooled odds ratios (ORs) with 95% confidence intervals (CIs) were calculated for three different models: allelic (X vs. x), dominant (XX and Xx vs. xx), and recessive (XX vs. Xx and xx). The random effect or fixed model was used depending on calculated heterogeneity level. Between-study heterogeneity and *I*^2^ were assessed by Cochrane’s Q test (significant heterogeneity was defined as *I*^2^ > 25%). A sensitivity analysis was performed to estimate the stability of the result. Publication bias was determined by Egger’s test.

## 3. Results

### 3.1. Study Selection

Figure 3 shows the flow diagram depicting the process of selecting studies for systematic review and the search strategy. A total of 298 items were found after a search of four databases. In all, 115 of these items were excluded as duplicates. After assessing their titles and abstracts, a further 140 articles were excluded as inaccurate/unrelated to the subject of the study. Finally, 43 articles were fully reviewed for eligibility. Of these, 33 were excluded for different reasons, as shown in Figure 3. Ten studies met the inclusion criteria and are presented below [32,33,34,35,36,37,38,39,40,41].

### 3.2. Study Characteristic

Of the ten studies included, seven assessed the *VDR* gene, while other genes studied were *CYP2R1*—cytochrome P450 family 2 subfamily R member 1 (two studies), *CYP24A1*—cytochrome P450 family 24 subfamily A member 1 (two studies), *CYP27B1*—cytochrome P450 family 27 subfamily B member 1, *DHCR7*—7-dehydrocholesterol reductase, and *CUBN*—cubilin (each gene assessed by one study). As expected, most of these studies (all but one) analyzed patients with a diagnosis of DTC. Only in one study were MTC patients enrolled, while there are no data about the anaplastic thyroid cancer [32,33,34,35,36,37,38,39,40,41]. Detailed characteristics of the studies included are presented in Table 1.

### 3.3. Study Results

#### 3.3.1. *VDR*

*VDR* gene SNPs have been studied for the first time in DTC patients by Haghpanah et al. in 2007 (five SNPs: rs2228570/*Fok*I, rs1544410/*Bsm*I, rs7975232/*Apa*I, rs731236/*Taq*I, and rs757343/*Tru9*I). No significant differences in genotype distributions were observed between patients and controls (only a difference in rs757343 allele frequencies between groups) [32]. In a study from 2009, Penna-Martinez et al. found that AA and Aa genotypes of the rs7975232 *VDR* SNP, FF of the rs2228570 SNP, and tABF haplotype are associated with a decreased risk of FTC. Instead, the Tabf haplotype may increase the risk of FTC. No associations were observed for PTC in this study (no differences found in both genotype and haplotype distributions). There were also no significant associations between vitamin D level or lymphocytic infiltration of thyroid gland and the SNPs studied [33]. On the contrary, Beysel et al. reported that TT and CT (ff and Ff) genotypes of rs2228570 *VDR* SNP may confer an increased risk of PTC. Moreover, these variants of rs2228570 SNP correlated with a more aggressive course of the disease including a more advanced stage, larger tumor size, more frequent lymph node involvement, extrathyroidal invasion, multifocality, and need for radioiodine treatment [37]. The authors did not find any associations for other studied *VDR* polymorphisms (rs7975232, rs1544410 and rs731236) [37]. Another Turkish study from 2021 did not confirm these results as there were no significant associations between any of the four *VDR* SNPs and DTC. The only significant association was between FF (CC) genotype and nodular goiter in comparison to healthy controls [40]. Lushchyk et al. studied additionally another *VDR* SNP—rs11568820/Cdx2. On a small DTC patients’ group, they showed its association with increased disease risk. No association was found with other SNPs studied (rs7975232, rs1544410 and rs731236) [36]. In the latest study of the role of *VDR* polymorphism in the development of DTC, again a set of four SNPs was analyzed: rs2228570, rs7975232, rs1544410, and rs731236. The authors showed a potential predisposing role of Ff and FF genotypes of rs2228570 SNP and Tt heterozygote of rs731236 SNP in thyroid cancer pathogenesis [41].

In only one study, *VDR* gene SNPs were assessed in MTC patients. Among the three SNPs studied (rs2228570, rs1544410 and rs757343), a significant association with cancer risk was found only for rs757343 SNP. Tt and tt genotypes were not only a risk factor for developing MTC but were also associated with a more aggressive course of the disease (more advanced stage, presence of metastases). Tt and tt genotypes of rs757343 also correlated with a higher 25(OH)D concentration, but only in the patients’ group [39]. All study results are also presented in Table 2.

#### 3.3.2. *CYP2R1*

Penna-Martinez et al. studied two *CYP2R1* SNPs in DTC (rs12794714, rs10741657), although no significant differences in allele or genotype frequencies were observed between the patients and controls [34]. In another study by Carvalho et al., rs2060793 SNP of the *CYP2R1* gene was tested for its association with the risk of DTC. However, again no significant association was found [38]. There were also no significant associations between the studied SNPs and other parameters in both papers.

#### 3.3.3. *CYP24A1*

In a study by Penna-Martinez et al., none of the polymorphisms of *CYP24A1* studied (rs927650, rs2248137 and rs2296241) were found to confer an increased risk of thyroid cancer development. There was only a small difference (not significant after adjustment) in allele distribution of rs2296241 SNP between FTC and healthy controls. However, further haplotype distribution analyses revealed some significant differences, mainly between the FTC group and controls [34]. The authors also observed that some haplotype combinations of *CYP24A1* may be associated with a lower level of circulating 1,25(OH)2D3 [34]. Carvalho et al. assessed another *CYP24A1* SNP-rs6013897, although they did not observe significant difference between DTC patients and controls [38].

#### 3.3.4. *CYP27B1*

In 2012, Penna-Martinez et al. tested the role of two *CYP27B1* polymorphisms (rs10877012, rs4646536) in the pathogenesis of DTC. They failed to find any significant differences between patients (PTC or FTC) and controls in genotype and allele distribution. Haplotype analysis also did not show any evident influence on disease risk or vitamin D level [34].

#### 3.3.5. *DHCR7*

In a study by Carvalho et al., it was found that G allele and GG/TG genotypes of *DHCR7* rs12785878 SNP may be associated with an increased overall risk of DTC (both PTC and FTC). No influence on clinicopathologic parameters was found [38]. No other studies have tested rs12785878 polymorphism in thyroid cancer.

#### 3.3.6. *CUBN*

Cubilin encoded by *CUBN* gene was found to be expressed on thyrocytes as well. One of the SNPs of *CUBN*—rs1801222 was studied for its association with DTC risk (PTC and FTC cases). However, no significant differences in genotype distribution were found between patients and controls. There was only some increase in PTC risk in the case of a combination of G allele and vitamin D deficiency (35).

### 3.4. Meta-Analysis

A total of seven studies assessing *VDR* SNPs in thyroid cancer were found. Six different SNPs were involved: rs2228570, rs7975232, rs1544410, rs731236, rs757343, and rs11568820 [32,33,36,37,39,40,41]. In the case of one study, genotype and allele frequencies were not available [36]. In a paper by Ramezani et al., MTC was analyzed, contrary to other studies assessing DTC patients (PTC, FTC, or DTC without distinguished subtypes) [39]. Moreover, in the case of two SNPs, rs757343 and rs11568820, the number of studies presenting results was too low (two and one, respectively) [32,36,39]. Therefore, finally five different papers were included in the meta-analysis, with data on four different *VDR* SNPs (rs2228570, rs7975232, rs1544410, rs731236) [32,33,37,40,41].

The results of meta-analysis are presented in Table 3. None of the SNPs studied showed any significant associations. Sensitivity analysis was performed, showing that overall ORs did not change significantly after single study elimination. To assess the publication bias, Egger’s test was used, and in two analyses *p* was less than 0.05 (rs7975232 recessive, rs1544410 recessive model). Heterogeneity analysis results are shown in Table 3.

## 4. Discussion

Vitamin D may directly affect the thyroid gland, as thyrocytes express *VDR*. Numerous studies have shown that the VDR level is increased in the case of DTC (mainly PTC was assessed) in comparison to the normal thyroid [18,42,43,44,45,46]. Moreover, *VDR* was observed among genes overexpressed in the follicular variant of PTC comparing to follicular adenoma [47]. Choi et al. found that high VDR level was associated with a more advanced stage of the disease, an increased risk of lateral neck lymph node metastases, and shorter recurrence-free survival [42]. In contrast, Clinckspoor et al. observed that *VDR* expression was lower in the case of lymph node metastases [45]. Unlike DTC, in many anaplastic thyroid cancer cases, loss of *VDR* expression was reported [45]. Observations of other neoplasms also show that loss of cell differentiation is often associated with reduced *VDR* expression. In the development of cancer, both genetic (mutations, polymorphisms) and epigenetic modifications occur that can lead to changes in the VDR protein level or its function [48,49]. The association between *VDR* SNPs and cancer risk was shown among others in the case of breast, keratinocyte, colorectal ovarian, or lung malignancies [50,51,52]. Unfortunately, in the case of thyroid cancer, the results of papers published to date seem heterogeneous and largely inconclusive (as mentioned above). Rs2228570 *VDR* SNP, a polymorphism with a proven functional role, is suggested by some researchers as modulating the risk of disease development [33,37,41]. However, a meta-analysis of available data did not confirm a significant association between rs2228570 *VDR* SNP or the other three most frequently studied *VDR* SNPs (rs1544410, rs7975232, rs731236) and DTC risk. Further studies and larger groups of patients are needed.

Less is known about the potential role of the *CYP2R1* gene, encoding a major enzyme with vitamin D 25-hydroxylase activity, in neoplasm development. However, some studies have shown an association between polymorphisms of *CYP2R1* and cancer risk (e.g., breast) [53] or suggested altered gene expression in tumor cells (e.g., renal cell carcinoma) [54]. CYP2R1 mRNA was detected in both normal thyrocytes and various thyroid cancer subtypes (including anaplastic, papillary, and follicular thyroid cancer). Gene expression was observed to be higher in thyroid cancer cell lines, although these differences were only modest [55]. Among the three different SNPs of *CYP2R1* studied so far in DTC patients, there were no significant associations with disease risk [34,38].

25-hydroxyvitamin D-1 alpha hydroxylase and its gene-*CYP27B1* have been broadly studied for association with different tumors susceptibility. Significant associations between *CYP27B1* expression or gene polymorphisms and cancer risk were found, for example, for colorectal, ovarian, and non-small cell lung cancer [56,57,58,59]. Both normal thyroid follicular cells and DTC cells express *CYP27B1*. Some authors postulate that its level is increased in the case of DTC cells [44,45,55]. In contrast, Yavropoulou et al. and Balla et al. did not observe significant differences in *CYP27B1* expression between PTC and non-neoplastic thyroid cells [43,60]. In anaplastic thyroid cancer, there was a tendency to more frequent negative results for CYP27B1 with increasing Ki67 or in cases with distant metastases [45]. However, the role of two studied *CYP27B1* polymorphisms (rs10877012, rs4646536) in the pathogenesis of DTC was not confirmed [34].

*CYP24A1* expression has been observed to be increased in the course of different malignancies e.g., colon, ovarian, lung, liver, or esophageal cancer [60]. Increased activity of 24-hydroxylase in the tumor tissues could be a mechanism for the neutralization of vitamin D antitumor activity. Most of the studies of DTC also confirmed that *CYP24A1* expression is increased in cancerous compared to normal cells [43,45,60,61,62]. There was also an association between higher *CYP24A1* expression and resistance to active vitamin D analogs [63]. Yavropoulou et al. observed that there is an association between *CYP24A1* expression and the risk of lymph node metastases or extrathyroidal extension of the tumor [43]. In other studies, a positive correlation between *CYP24A1* expression and the risk of vascular invasion, lymph node metastasis, tumor size [60], or more advanced stage of the disease [62] was observed. A study on anaplastic thyroid cancer cell lines showed that *CYP24A1* knockdown led to the enhancement of vitamin D antitumor activity [64]. Another approach is to search for polymorphisms that may impair gene function and, in consequence, facilitate the disease development. There are suggestions about the association between *CYP24A1* SNPs and some cancer types, including prostate, breast, or pancreas cancer [65,66]. However, in the case of thyroid, only a weak association between rs2296241 and FTC has been observed [34]. Rs2296241 is a synonymous SNP, although it could influence gene expression or protein function in other mechanisms or be in linkage disequilibrium with different, functional, SNPs [67].

Among the other vitamin D-related genes, there are also some limited data about *DHCR7* role in DTC. 7-dehydrocholesterol reductase is postulated to regulate the balance between cholesterol and vitamin D synthesis [68]. Carvalho et al. found that the G allele and GG/TG genotypes of rs12785878 SNP may be associated with an increased overall risk of DTC (both PTC and FTC). No influence on clinicopathologic parameters was found [38]. Unfortunately, no other studies have yet searched for the role of this gene in thyroid gland neoplasms. *DHCR7* is one of a few genes found to influence 25(OH)D plasma level in GWAS [69]. Rs12785878 SNP was also found to increase a general cancer risk in a meta-analysis of five studies including patients with breast, hepatocellular, and thyroid cancer [69].

Cubilin encoded by the *CUBN* gene plays variable roles, including the process of renal 1,25(OH)2D synthesis (25(OH)D—vitamin D binding protein (DBP) complexes absorption from glomerular ultrafiltrate) [70]. It was found to be expressed on thyrocytes too. One of the SNPs of *CUBN*-rs1801222- was studied for its association with DTC, although no significant differences were found between patients and controls (35).

There are also other candidates in the group of vitamin D-related genes that have not yet been studied in thyroid cancer patients. Among them, there is *CYP27A1* (cytochrome P450 family 27 subfamily A member 1) encoding sterol 26-hydroxylase, an enzyme playing a minor role as vitamin D 25-hydroxylase. *CYP27A1* was found to be expressed in both thyroid cancer cells and normal thyrocytes [55]. In addition to classic vitamin D metabolism pathways, also alternative routes have been reported. One is mediated by cholesterol side-chain cleavage enzyme encoded by *CYP11A1* (cytochrome P450 family 11 subfamily A member 1; enzyme expressed, i.e., in the gastrointestinal tract or skin, catalyzes hydroxylation predominantly at C-20 and C-22) [71]. The main product of its action-17,20,23(OH)3D, promotes differentiation and inhibits proliferation and inflammatory processes. It was found to work as a biased noncalcemic agonist of VDR with possible anticancer activity [71,72]. Similarly, the *GC* gene that encodes DBP requires further exploration. This protein, which is responsible among other things for vitamin D and its metabolites transportation, was significantly under-expressed in the samples of papillary thyroid cancer comparing to normal thyroid cells [73,74]. In a recent study by Mull et al., an inverse correlation was observed between *DBP* expression in thyroid cancer tissue samples and PTC staging [75].

## 5. Conclusions

There are relatively large and comprehensive reports on the role of vitamin D in the pathogenesis of thyroid neoplasms. Some in vitro studies using vitamin D or its analogs also seem promising. It is expected that vitamin D-related gene variants could also have an important modulating role in thyroid tumorigenesis. To date, polymorphic variants of *VDR* have been studied most extensively, although results are inconclusive. In the case of other vitamin D-related genes, only single studies are available or even no specific results. Therefore, further effort is needed to explain the complex interaction between vitamin D concentration, vitamin D-related gene variants modifying its metabolism, and thyroid cancer pathogenesis.

## Figures and Tables

**Figure 1 ijms-23-13661-f001:**
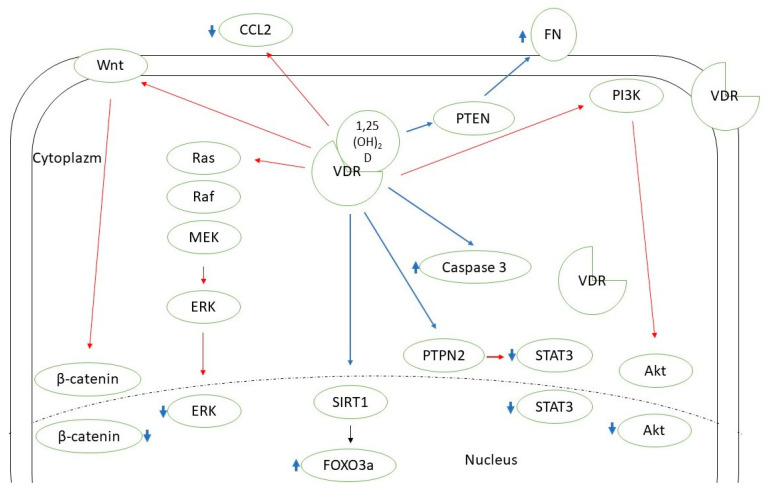
Known pathways mediating vitamin D action in thyroid cancer cells. CCL2—C-C motif chemokine ligand 2; ERK—extracellular signal-regulated kinase; FN—fibronectin; FOXO3a—forkhead box protein O3a; MEK—mitogen-activated protein kinase kinase 1; PI3K—phosphoinositide 3-kinase; PTEN—phosphatase and tensin homolog; PTPN2—protein tyrosine phosphatase N 2; SIRT1—sirtuin 1 histon deacethylase; STAT3—signal transducer and activator of transcription 3; VDR—vitamin D receptor. Red lines—inhibition; blue lines—stimulation.

**Figure 2 ijms-23-13661-f002:**
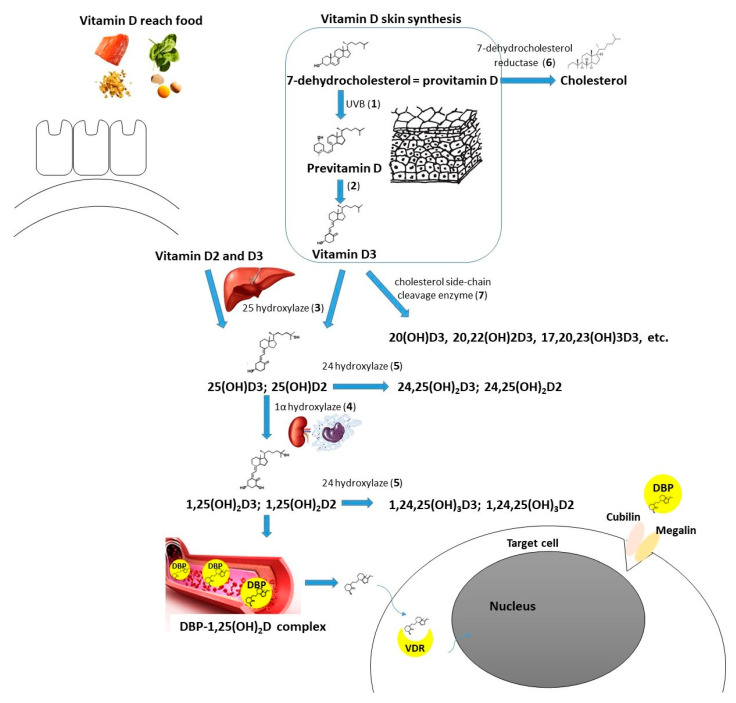
Vitamin D metabolism (including molecules/enzymes involved, coding genes in brackets): Cubilin (*CUBN*); DBP—vitamin D binding protein (*GC*); Megalin (*LRP2*); VDR—vitamin D receptor (*VDR*); 1—UVB radiation, 290–315 mm wavelength; 2—nonenzymatic isomerization reaction under the temperature; 3—25-hydroxylase (main enzyme—*CYP2R1*—cytochrome P450 family 2 subfamily R member 1, minor role of *CYP27A1*—cytochrome P450 family 27 subfamily A member 1); 4—1α-hydroxylase (*CYP27B1*—cytochrome P450 family 27 subfamily B member 1); 5—24-hydroxylase (*CYP24A1*—cytochrome P450 family 24 subfamily A member 1); 6—7-dehydrocholesterol reductase (*DHCR7*); 7—cholesterol side-chain cleavage enzyme (*CYP11A1—*cytochrome P450 family 11 subfamily A member 1).

**Figure 3 ijms-23-13661-f003:**
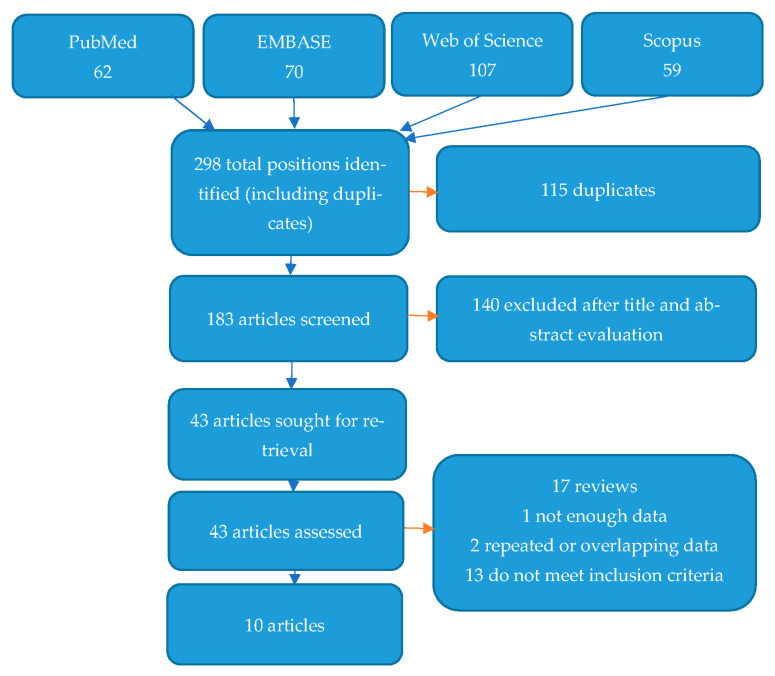
Flow diagram demonstrating the search and selection process for articles.

**Table 1 ijms-23-13661-t001:** Characteristics of the studies included.

Author	Year	Country	Ethnicity	Patients			Controls			Genes Studied	Method		Other Parameters Measured
				N	Age (Mean ± SD)	Gender (F, %)	N	Age (Mean ± SD)	Gender (F, %)	Type of Control			
Haghpanah V. [32]	2007	Iran	Caucasian	71 DTC	44.82 ± 12.60	59 (83.10)	82	43.93 ± 13.30	N/A	Patients hospitalized for other ailments, negative malignancy history	*VDR*	PCR-RFLP	Not studied
				58 PTC	43.73 ± 12.81	47 (81.03)							
				13 FTC	49.69 ± 10.76	12 (92.31)							
Penna- Martinez M. [33]	2009	Germany	Caucasian	172 DTC	53.23 ± 15.73	111 (64.53)	321	47.4 ± 12	135 (42.06)	Blood donors (medical staff and students), negative family history	*VDR*	PCR-RFLP	25(OH)D, 1,25(OH)2D, antithyroid antibodies, lymphocytic infiltration
				132 PTC	51.9 ± 15.4	86 (65.15)							
				40 FTC	57.6 ± 16.2	25 (62.50)							
Penna- Martinez M. [35]	2012	Germany	Caucasian	230 DTC	N/A	153 (66.52)	294	N/A	130 (44.22)	N/A	*CUBN*	probe-based quantitative PCR	25(OH)D, 1,25(OH)2D
				186 PTC									
				44 FTC									
Penna- Martinez M. [34]	2012	Germany	Caucasian	253 DTC	55 (MED)	167 (66.01)	302	38 (MED)	138 (45.70)	Blood donors (medical staff and students), negative family history	*CYP2R1* *CYP24A1* *CYP27B1*	PCR-RFLP or probe-based quantitative PCR	25(OH)D, 1,25(OH)2D
				205 PTC	N/A	137 (66.83)							
				48 FTC	N/A	30 (62.50)							
Lushchyk M. [36]	2018	Belarus	Caucasian	26 DTC	N/A	N/A	64	N/A	N/A	N/A	*VDR*	probe-based quantitative PCR	25(OH)D
Beysel S. [37]	2018	Turkey	Caucasian	165 PTC	46.89 ± 13.22	134 (81.2)	172	45.25 ± 4.89	149 (86.8)	No details, thyroid and other serious diseases excluded	*VDR*	probe-based quantitative PCR	25(OH)D, clinical features of thyroid cancer
Carvalho I.S. [38]	2019	Portugal	Caucasian	500 DTC	46.1 ± 13.9	409 (81.80)	500	35.9 ± 14.2	387 (77.40)	Blood donors	*DHCR7* *CYP2R1 CYP24A1*	PCR-RFLP	Clinical features of thyroid cancer
				442 PTC	N/A	N/A							
				58 FTC									
Ramezani M. [39]	2020	Iran	Caucasian	40 MTC	36 ± 7.57	26 (65.00)	40	33 ± 6.02	N/A	Healthy medical staff members, negative family history	*VDR*	Direct sequencing	25(OH)D
Gunes A. [40]	2021	Turkey	Caucasian	103 DTC	48.90 ± 14.40	82 (79.61)	216	51.32 ± 13.1	168 (77.78)	Outpatients with no thyroid pathologies	*VDR*	PCR-RFLP	Not studied
				113 BN	49.39 ± 12.61	86 (76.11)							
Cocolos A.M. [41]	2022	Romania	Caucasian	113 DTC	50 ± 14.46	91 (80.56)	150	55.87 ± 11.67	137 (91.33)	Patients with benign thyroid diseases	*VDR*	PCR-RFLP	25(OH)D, ultrasound features

BN—benign nodules; CUBN—cubilin; CYP24A1—cytochrome P450 family 24 subfamily A member 1; CYP27B1—cytochrome P450 family 27 subfamily B member 1; CYP2R1—cytochrome P450 family 2 subfamily R member 1; DHCR7—7-dehydrocholesterol reductase; DTC—differentiated thyroid cancer; F—female; SD—standard deviation; FTC—follicular thyroid cancer; MED—median; MTC—medullary thyroid cancer; N—number of patients/controls; N/A—not available; PCR—polymerase chain reaction; PTC—papillary thyroid cancer; RFLP—restriction fragments length polymorphism; SNP—single nucleotide polymorphism; VDR—vitamin D receptor.

**Table 2 ijms-23-13661-t002:** Association between vitamin D-related gene polymorphisms and thyroid cancer—results of the studies.

Gene	Author	Sample Size (Patients/ Controls)	Studied SNP	Association (+/−)			Disease Course (+/−)	Other Associations Studied
				Genotypic	Allelic	Haplotypes		
*VDR*	Haghpanah V. 2007 [32]	71 DTC/82	rs2228570*/Fok*I	−	−	Not studied	Not studied	Not studied
			rs7975232*/Apa*I	−	−			
			rs1544410*/Bsm*I	−	−			
			rs731236*/Taq*I	−	−			
			rs757343*/Tru9*I	−	+			
	Penna-Martinez M. 2009 [33]	172 DTC/321	rs2228570*/Fok*I	+ FTC only	Not studied	+	Not studied	25(OH)D and 1,25(OH)2D level—no association
			rs7975232*/Apa*I	+ FTC only				
			rs1544410*/Bsm*I	−				
			rs731236*/Taq*I	−				
	Lushchyk M. 2018 [36]	26 DTC/64	rs7975232*/Apa*I	−			Not studied	Not studied
			rs1544410*/Bsm*I	−				
			rs731236*/Taq*I	−				
			rs11568820/ Cdx2	+				
	Beysel S. 2018 [37]	165 PTC/ 172	rs2228570*/Fok*I	+	Not studied	Not studied (SNPs not in LD)	+ (stage of the disease, multifocality, lymph node involvement)	Not studied
			rs7975232*/Apa*I	−			−	
			rs1544410*/Bsm*I	−			−	
			rs731236*/Taq*I	−			−	
	Ramezani M. 2020 [39]	40 MTC/ 40	rs2228570*/Fok*I	−	−	Not studied	−	−
			rs1544410*/Bsm*I	−	−		−	−
			rs757343*/Tru9*I	+	+		+ (aggressiveness of the disease)	+ (vitamin D level)
	Gunes A. 2021 [40]	103 DTC, 113 BN/216	rs2228570*/Fok*I	+ BN only	Not studied	Not studied	Not studied	Not studied
			rs7975232*/Apa*I	−				
			rs1544410*/Bsm*I	−				
			rs731236*/Taq*I	−				
	Cocolos A.M. 2022 [41]	113 DTC/137	rs2228570*/Fok*I	+	Not studied	Not studied	+ (stage of the disease, multifocality, cervical lymph node metastases)	Not studied
			rs7975232*/Apa*I	−			−	
			rs1544410*/Bsm*I	−			−	
			rs731236*/Taq*I	+			−	
*CYP27B1*	Penna-Martinez M. 2012 [34]	253 DTC/302	s10877012	−	−	−	Not studied	25(OH)D and 1,25(OH)2D level—no association
			rs4646536	−	−			
*CYP24A1*	Penna-Martinez M. 2012 [34]	253 DTC/302	rs927650	−	−	+	Not studied	+ (1,25(OH)2D status)
			rs2248137	−	−			
			rs2296241	+ FTC (weak)	+ FTC (weak)			
	Carvalho I.S. 2019 [38]	500 DTC/500	rs6013897	−	−	Not studied	−	Not studied
*DHCR7*	Carvalho I.S. 2019 [38]	500 DTC/500	rs12785878	+	+	Not studied	−	Not studied
*CYP2R1*	Penna-Martinez M. 2012 [34]	253 DTC/302	rs12794714	−	−	−	Not studied	25(OH)D and 1,25(OH)2D level—no association
			rs10741657	−	−			
	Carvalho I.S. 2019 [38]	500 DTC/500	rs2060793	−	−	Not studied	+ (risk of metastasis-weak)	Not studied
*CUBN*	Penna-Martinez M. 2012 [35]	230 DTC/294	rs1801222	−	N/A	Not studied	Not studied	− (no evident influence on vitamin D status)

BN—benign nodules; CUBN—cubilin; CYP24A1—cytochrome P450 family 24 subfamily A member 1; CYP27B1—cytochrome P450 family 27 subfamily B member 1; CYP2R1—cytochrome P450 family 2 subfamily R member 1; DHCR7—7-dehydrocholesterol reductase; DTC—differentiated thyroid cancer; FTC—follicular thyroid cancer; LD—linkage disequilibrium; MTC—medullary thyroid cancer; N/A—not available; PTC—papillary thyroid cancer; SNP—single nucleotide polymorphism; VDR—vitamin D receptor.

**Table 3 ijms-23-13661-t003:** Meta-analysis results—association between four vitamin D receptor gene polymorphisms (rs2228570, rs1544410, rs7975232 and rs731236) and differentiated thyroid cancer risk.

Comparison Model	Sample Size (DTC/Controls)	*I*^2^ (%)	*p* ^H^	Effect Model	OR (95% CI)	*p* ^Z^
rs2228570/*Fok*I						
Allelic	647/860	86	<0.01	Random	1.11 (0.74–1.68)	0.60
Recessive		57	0.05	Random	0.93 (0.67–1.30)	0.68
Dominant		86	<0.01	Random	1.39 (0.64–2.99)	0.40
rs1544410/*Bsm*I						
Allelic	624/825	48	0.10	Random	1.05 (0.85–1.31)	0.63
Recessive		37	0.17	Random	1.05 (0.76–1.47)	0.76
Dominant		21	0.28	Fixed	1.05 (0.83–1.34)	0.68
rs7975232/*Apa*I						
Allelic	624/828	29	0.23	Random	1.16 (0.97–1.39)	0.10
Recessive		0	0.55	Fixed	1.13 (0.90–1.41)	0.29
Dominant		37	0.17	Random	1.34 (0.94–1.90)	0.11
rs731236/*Taq*I						
Allelic	624/827	74	<0.01	Random	1.01 (0.75–1.38)	0.93
Recessive		0	0.85	Fixed	0.88 (0.71–1.10)	0.26
Dominant		88	<0.01	Random	1.16 (0.49–2.74)	0.73

*p*^H^—*p* value of heterogeneity test; *p*^Z^—*p* value of Z test. DTC—differentiated thyroid cancer; OR—odds ratio; CI—confidence interval.

## Data Availability

Not applicable.

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
