# Peer review of "Vitamin D-Related Genes and Thyroid Cancer—A Systematic Review"

_ijms, 2022, doi:10.3390/ijms232113661_

Round 1

Reviewer 1 Report

The paper is ready to publish

Author Response

Thank you for your interest and positive opinion. Best regards

Reviewer 2 Report

The article is interesting and worth publishing, with the suggested major corrections. Unfortunately, throughout the article, the authors did not follow the requirements for citing references in the manner suggested, in the instructions for authors. The authors did not include a Conclusions section, so this required section should be completed. On the plus side, the article has numerous cited foreign literature, but unfortunately incorrectly cited in the References section.

Author Response

Thank you for your remarks and recommendations. The conclusions section has been added and some changes have been made in this part of the manuscript  (including also other reviewer comments). References are not fully in agreement with the Journal guidelines. However, IJMS accepts free format submissions and also references in different styles are acceptable on submission. Moreover, this format is easier to modify when still some changes are made according to reviewers' reports. The reference format will be adjusted after acceptance. 

Reviewer 3 Report

The review carried out by the authors is interesting, but perhaps it is not yet the appropriate time to carry out a systematic review due to the few published works on the subject.

The authors finally develop the content of only 10 articles.

Some major issues should be considered:

1. Both the objective and the conclusion should be more focused on the results of the review.

2. In the introduction, many statements are made that are not verified with any bibliographical reference.

3. Table 1, the abbreviations of all the acronyms used in the table, should be included.

4. Authors should use the "rs" nomenclature to define all analysed SNPs. In addition, they should indicate which is the wild-type or mutant allele for each polymorphism and its MAF.

5. Many polymorphisms are not defined by the "rs" nomenclature, indicating only the restriction enzyme involved in molecular cleavage.

6. Throughout the text, the authors tend to repeat the bibliographic citation, try to write the text in another way to avoid duplicity.

7. Many genes are abbreviated without indicating their involvement in the topic.

8. Table 2. Authors should show the genes on the left of the table, as they show them is confusing and hard to read. The references do not have the year of publication.

9- Table 3. pH or Ph??

Author Response

  1. Both the objective and the conclusion should be more focused on the results of the review.

Some changes have been made in mentioned parts of the manuscript to adjust to the suggestions.

  1. In the introduction, many statements are made that are not verified with any bibliographical reference.

Some additional references have been added in the introduction section.

  1. Table 1, the abbreviations of all the acronyms used in the table, should be included.

All missing abbreviations have been added. I was not sure if there is a need to explain also the gene acronyms as they are official gene symbols and should generally be unequivocal.

  1. Authors should use the "rs" nomenclature to define all analyzed SNPs. In addition, they should indicate which is the wild-type or mutant allele for each polymorphism and its MAF.
  2. Many polymorphisms are not defined by the "rs" nomenclature, indicating only the restriction enzyme involved in molecular cleavage.

Rs nomenclature has been added to the text and tables. In the case of VDR polymorphisms restriction enzyme-based names are frequently used in the literature. However, you are right that it would be better to use rs names as it is more precise and clear. Both versions are now available in the case of first use in the text and in tables.

Wild-type or mutant allele for each polymorphism and different SNPs MAFs are usually described in references. Allele frequencies and MAF values vary depending on the population studied with ethnic differences as for example in the case of rs11568820 VDR SNP where the opposite allele can be dominant depending on population. Do you mean each study MAFs or “global MAF” as calculated for example by 1000Genome project? If you think it is necessary I can try to add this element for clarity of the interpretation of described results.   

  1. Throughout the text, the authors tend to repeat the bibliographic citation, try to write the text in another way to avoid duplicity.

Some modifications have been made to these elements of the manuscript.

  1. Many genes are abbreviated without indicating their involvement in the topic.

Figure 1 (renamed to Fig. 2) has been modified to show the involvement or role of some other genes. Genes described have been selected taking into account previous publications on the role of vitamin D metabolism in different diseases. Some additional comments have been added in the text to describe the role of, for example, DHCR7 or CUBN genes.

  1. Table 2. Authors should show the genes on the left of the table, as they show them is confusing and hard to read. The references do not have the year of publication.

Some modifications have been made. I hope the table is now more clear. The year of publications was added.

9- Table 3. pH or Ph??

The correct version should be (changed in the manuscript) pH or p heterogeneity to distinguish from pZ.

Reviewer 4 Report

General comment:

This manuscript, entitled “Vitamin D-related genes and thyroid cancer – a systematic review,” authored by Maciejewski and Lacka, systematically reviewed the role of Vit D-related genes in thyroid cancer. A further meta-analysis of the polymorphic effect of these genes on cancer added a layer of concise study. A descriptive survey like this helps target these proteins to combat the progression and pathogenicity of cancer diseases. This survey will provide essential links between specific cancer and different genes/proteins involved in vitamin D metabolism. Still, it will also inspire therapeutic approaches to control the progression of cancers. In my opinion, this is a valuable work and is suitable for publication in the International Journal of Molecular Sciences after the authors have addressed the following comments and questions:

Specific comments:

1)     Line 54-55 – missing reference

2)     Do you think cyp11a1 is missing in this vitamin d related p450 –

Ref 1 https://doi.org/10.1016/j.jsbmb.2013.10.012

Ref 2 https://doi.org/10.1016/j.freeradbiomed.2020.05.016

Ref 3 https://doi.org/10.1073/pnas.2336107100

3)     Please add a figure in detail for Vitamin D-regulated pathways in thyroid cancer.

4)     Is the relation of vit D to thyroid cancer very specific, or is it a common relation with any cancer?

5)     The mitochondrial calcium (Ca2+) fluxes are the key regulator for cardiac function – what is the associate proteins mutation's role here?

Author Response

  1. Reference has been added to the text as it was omitted by accident.
  2. Cyp11a1 was mentioned in the initial version of the manuscript and later removed to reduce the volume, as there is no direct information about the role of CYP11A1 polymorphisms in thyroid cancer etc. However, I agree that it is also potentially an important member of p450 cytochromes superfamily in the context of neoplasms. Some information about CYP11A1 has been added.
  3. A figure has been added to facilitate understanding the potential impact of vitamin D and its metabolism on thyrocyte and thyroid cancer pathogenesis. I tried to give some general information, although these pathways and interplay between different factors are in fact more complex.
  4. The relation between vitamin D or vitamin D-related genes and thyroid cancer seems not to be unique but common for at least some tumor subgroups. It is an effect of vitamin D pro-differentiation and anti-proliferation effects. VDR is active in multiple tissues and it is a reason for broad vitamin D effects. However, as shown in the text,  there are direct proofs of the vitamin D action of thyrocytes, not only by extrapolations from other neoplasm studies.
  5. The mitochondrial calcium (Ca2+) fluxes are described as important mechanisms in some tissues. The disruption of this balance can be involved in some pathologies including cancer. However, there is no data on this mechanism's role in thyroid physiology and pathology. On the contrary, there is some information about the role of thyroid hormones (fT3 and fT4) in regulating calcium fluxes.  Mitochondrial calcium homeostasis is mainly regulated by channels/transporters. Less is known about the influence of vitamin D and its genes.  

Round 2

Reviewer 2 Report

The recommendations have been included in the revised version of the manuscript. I have no further additional comments. Thank you.

Reviewer 3 Report

The authors have improved the article according to recommendations allowing a more adequate and agile reading.